# Impact of long- and short-range fibre depletion on the cognitive deficits of fronto-temporal dementia

**Melissa Savard[1,2], Tharick A Pascoal[1,3], Stijn Servaes[1], Thijs Dhollander[4], Yasser Iturria-Medina[5], Min Su Kang[1], Paolo Vitali[6], Joseph Therriault[1], Sulantha Mathotaarachchi[1], Andrea Lessa Benedet[1], Serge Gauthier[1,6,7], Pedro Rosa-Neto[1,6,7]\*, On behalf of for the Frontotemporal Lobar Degeneration Neuroimaging Initiative**

[1]Translational Neuroimaging Laboratory, The McGill University Research Centre for Studies in Aging, Montreal, Canada; [2]Douglas Hospital Research Centre, McGill University, Montreal, Canada; [3]Department of Psychiatry and Neurology, University of Pittsburgh, Pittsburgh, United States; [4]Developmental Imaging, Murdoch Children's Research Institute, Melbourne, Australia; [5]Montreal Neurological Institute, McGill University, Montreal, Canada; [6]Department of Neurology and Neurosurgery, McGill University, Montreal, Canada; [7]Department of Psychiatry, McGill University, Montreal, Canada

**\*For correspondence:**
pedro.rosa@mcgill.ca

**Competing interest:** The authors declare that no competing interests exist.

**Abstract** Recent studies suggest a framework where white-matter (WM) atrophy plays an important role in fronto-temporal dementia (FTD) pathophysiology. However, these studies often overlook the fact that WM tracts bridging different brain regions may have different vulnerabilities to the disease and the relative contribution of grey-matter (GM) atrophy to this WM model, resulting in a less comprehensive understanding of the relationship between clinical symptoms and pathology. Using a common factor analysis to extract a semantic and an executive factor, we aimed to test the relative contribution of WM and GM of specific tracts in predicting cognition in the Frontotemporal Lobar Degeneration Neuroimaging Initiative (FTLDNI). We found that semantic symptoms were mainly dependent on short-range WM fibre disruption, while damage to long-range WM fibres was preferentially associated to executive dysfunction with the GM contribution to cognition being predominant for local processing. These results support the importance of the disruption of specific WM tracts to the core cognitive symptoms associated with FTD. As large-scale WM tracts, which are particularly vulnerable to vascular disease, were highly associated with executive dysfunction, our findings highlight the importance of controlling for risk factors associated with deep WM disease, such as vascular risk factors, in patients with FTD in order not to potentiate underlying executive dysfunction.

## Editor's evaluation

This study explores how the pathophysiology of frontotemporal dementia, a collection of younger-onset dementias, impacts grey and white mater brain integrity, and how such changes relate to discrete aspects of cognition. The authors used whole-brain fixed-based analysis, structural connectivity analysis of white matter tracts, alongside voxel-based morphometry of grey matter atrophy. Overall, semantic impairment was found to associate with relatively short-range white matter dysfunction, while executive dysfunction was related to long-range white matter fibres.

## Introduction

Fronto-temporal dementia (FTD) is the second most prevalent form of early onset dementia (*Bang et al., 2015*; *Cairns et al., 2007*). The misfolding and aggregation of proteins such as tau, TDP-43, FUS, or ubiquitin-positive proteins encompass nearly all cases of FTD (*Seelaar et al., 2011*). FTD clinical phenotype includes behavioural, executive, and language dysfunction without primary amnesia. The initial clinical manifestation of the disease characterizes FTD cases into three common variants: the behavioural (BV) and two distinct forms of primary progressive aphasias, the semantic (SV) and the progressive non-fluent aphasia (PNFA). As the disease progresses, both language and behavioural dysfunction may appear in all variants.

While FTD has long been considered a grey-matter (GM) disease, recent advance in diffusion MRI (dMRI) research has revealed that white matter (WM) is also much involved in the pathophysiology of the disease (*Zhang et al., 2009*; *McKenna et al., 2021*). GM atrophy is typically observed in the insula (*Muhtadie et al., 2021*) (all variants), the bilateral anterior cingulate and frontal lobe (BV) (*Lanata and Miller, 2016*), left anterior temporal lobe (SV) (*Williams et al., 2005*), and left premotor and inferior frontal cortex (PNFA) (*McMillan et al., 2004*). Widespread WM abnormalities have been observed in the uncinate fasciculus, superior frontal, inferior frontal and inferior fronto-occipital fasciculi, the corpus callosum and the cingulum bundle, with a large overlap amongst participants; see *Greaves and Rohrer, 2019*; *Meeter et al., 2017*; *Rohrer et al., 2010*, for recent reviews of MRI findings in FTD. In some mutation carriers, WM changes are detectable up to 30 years before symptoms onset (*Jiskoot et al., 2018*), strengthening the importance of considering WM alteration as part of the pathophysiology of FTD.

Although often studied separately, WM and GM impairments are not occurring in isolation from one another. Alteration from GM may propagate to WM and, reciprocally, WM damage may propagate to GM via Wallerian degeneration or retrograde degeneration (*Metzler-Baddeley et al., 2019*; *Villain et al., 2008*; *Villain et al., 2010*). The aforementioned constructs suggest a dynamical and interdependent relationship between GM and WM as determinants of cognitive symptoms in neurodegenerative conditions such as FTD. Despite strong evidence of isolated effects of both WM and GM disruptions on FTD, their relative contribution to the impairment of the different cognitive domains typically affected in patients with FTD is unknown. Nonetheless, a number of studies have related both GM and WM atrophy to discrete aspects of cognition in BV and SD including disinhibition (*Piguet et al., 2011*), moral reasoning (*Strikwerda-Brown et al., 2021*), and WM changes over time (*Lam et al., 2014*).

In the present study, we aim to clarify the relative contribution of different properties of WM fibres and GM to the cognitive impairment (semantic and executive) in FTD patients. Specifically, we used a WM fixel-based analysis (FBA) combined with a structural connectivity and GM voxel-based morphometry (VBM) analyses to (1) provide an improved characterization of the whole brain fibre density (FD) and fibre cross-section (FC) impairment across the variants, (2) investigate the relationship between the WM metrics and GM volume, (3) evaluate the relationship between WM metrics and cognition domains in patients, and (4) test the relative contribution of WM and GM of specific tracts in predicting cognition. We studied these associations across different WM tracts under the assumption that these associations vary depending on specific WM tracts characteristics. We found that semantic symptoms were mainly dependent on short-range WM fibre disruption, while damage to long-range WM fibres was preferentially associated to executive dysfunction with the GM contribution to cognition being predominant for local processing.

## Results

### Fibre loss in FTD variants

Variants of FTD all had extensive WM impairments compared to normal controls after correction for age, sex, and intracranial volume (ICV) (*Figure 1*). *Figure 2a–c* shows the streamline segments associated to significantly reduced FC and FD (FWE-corrected p-value < 0.05; colour coded by direction) for the BV, PNFA, and SV, respectively. Irrespective of the variant, reduced FC (*Figure 2a–c*; upper panels) could be observed in large associative fibres including the uncinate fasciculus, the inferior fronto-occipital fasciculus and the superior longitudinal fasciculus, cingulum, and corpus callosum. Despite a large common network, variant-specific differences could be noted in the bilateral anterior and medial

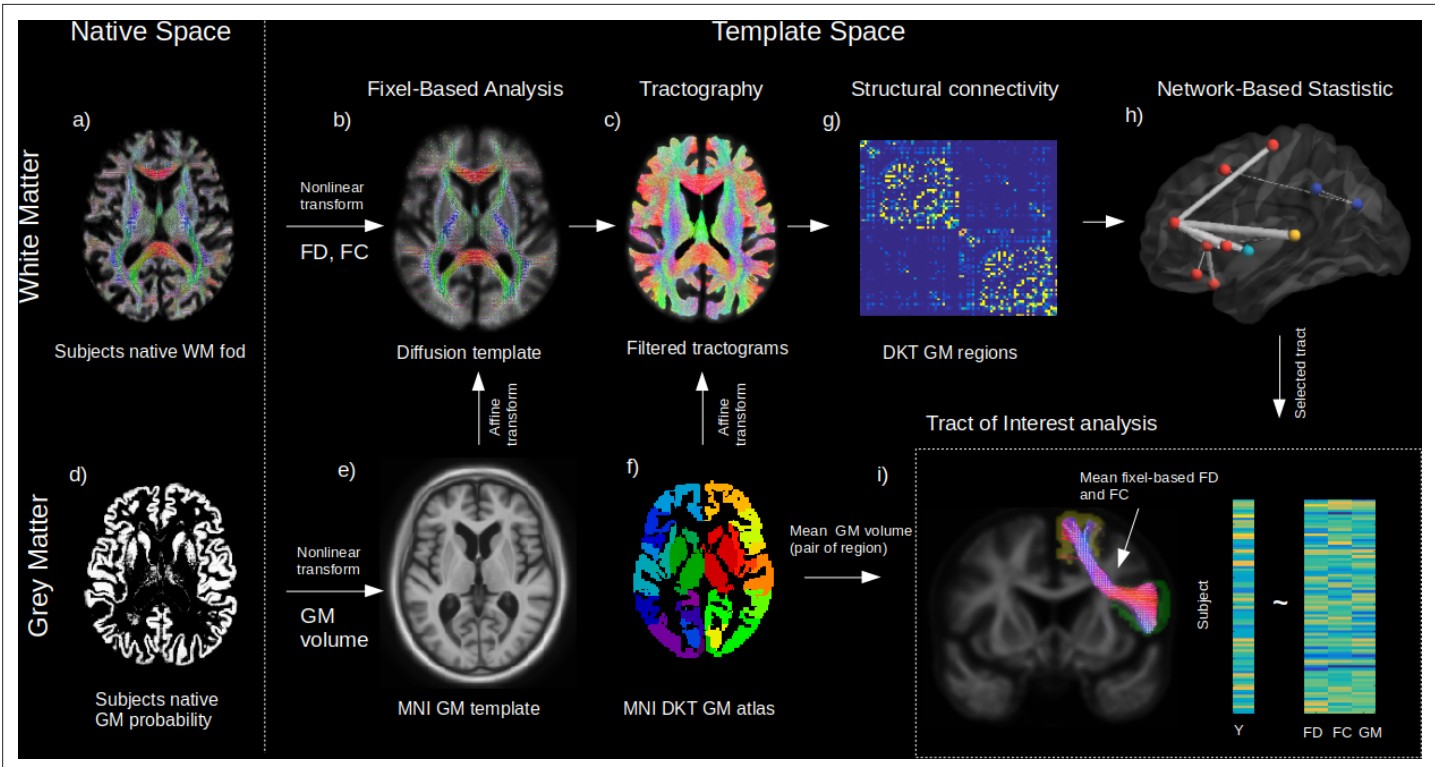

**Figure 1.** Method workflow. The main steps of the methods are shown from the native space (left) to template space (right). For the white matter (WM) (upper panels), native diffusion-weighted MRIs were first preprocessed to obtain individual normalized WM FODs (**a**). WM FODs were non-linearly registered to a study-specific WM FOD template (**b**), to obtain the fibre density (FD), and fibre cross-section metrics (FC), later used in whole brain fixel-based analysis. The template space WM FODs were then used to generate individual probabilistic tractograms (**c**). For the grey matter (GM) (lower panels), native space GM probability maps were warped to a study specific GM template in MNI space to obtain individual template space GM volume (**e**). An affine transform was estimated between MNI template and the diffusion template space which was subsequently applied to the Desikan-Killiany (DKT) GM atlas to bring the DKT atlas in diffusion space (**f**). Individual structural connectivity matrices were then obtained by counting the amount of fibres connecting each pair of GM regions within the DKT atlas (**g**). Significant difference in connectivity for a given dependant variable (Y) was then tested using the network-based statistic enhanced (**h**). Significant predictors (connections) were selected to access the relative importance of GM volume and WM (FD and FC) within each connection in predicting Y (**h**), where mean FD and FC were obtained in fixels belonging to the connection streamlines and GM was the average of both GM regions volume for each subject.

part of the frontal cortex and lateral orbitofrontal WM for BV (*Figure 2a*), while PNFA presented with reduced FC in the caudal part (precentral gyrus/supplementary motor area [SMA]) of the left frontal cortex (*Figure 2b*) and SV showed a left predominant FC reduction in the inferior longitudinal fasciculus (*Figure 2c*). Reduced FD patterns (*Figure 2a–c*; lower panels) were similar to those observed for FC although with a lower spatial extent.

Structural connectivity analysis (*Figure 2d–f*), although based on a different method (tractography), provided complementary information to the FBA, about specific GM regions that may be affected by the WM impairment. Significant reductions (FWE-corrected p-value < 0.001) in tracts connecting GM regions are shown for frontal regions (red), the insula (light blue), the temporal lobe (yellow), subcortical region (green), and parietal regions (dark blue), where the line thickness corresponds to the strength of the effect. Compared to normal controls, BV (*Figure 2d*) had the largest reduction in bilateral insula – inferior frontal cortex (pars opercularis and triangularis) connectivity, followed by bilateral reduction in thalamo-frontal (rostral middle frontal) connectivity. For PNFA (*Figure 2e*), the largest reduction was also observed in insula – inferior frontal cortex (pars opercularis and triangularis) connectivity but in the left hemisphere only, followed by precentral – middle frontal connectivity impairment. For SV (*Figure 2f*), the largest reduction occurred in the left hemisphere between the thalamus and the temporal cortex (superior and middle), but also between the lateral orbitofrontal and superior frontal cortex, followed by intra-temporal connectivity reduction.

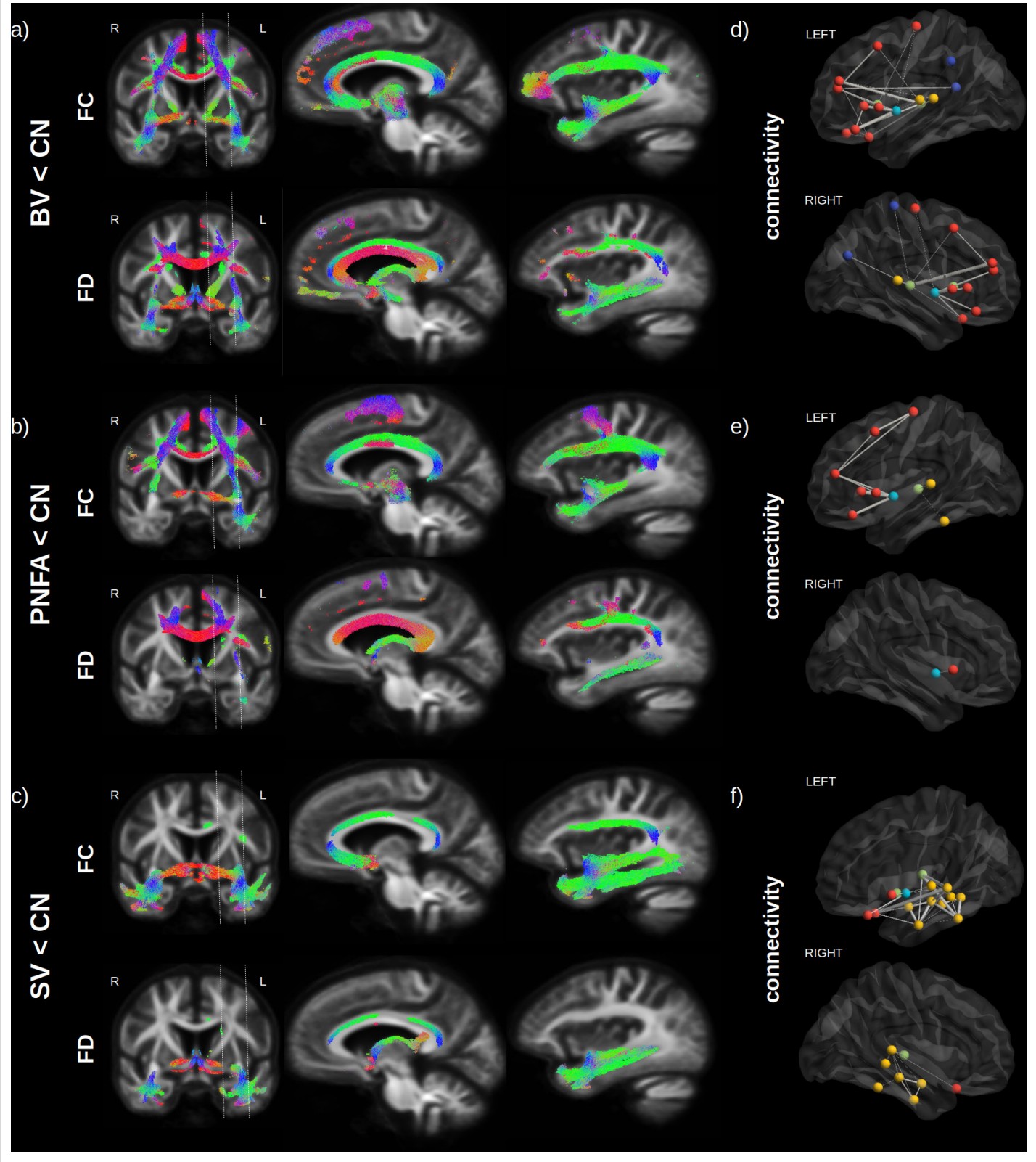

**Figure 2.** Fibre loss in fronto-temporal dementia (FTD) variants. Streamlines (colour coded by direction) associated to significantly reduced fibre cross-section (FC) and fibre density (FD) (FWE-corrected p-value) are shown for behavioural variants (BV) vs. normal elderly control (CN) (**a**), progressive non-fluent aphasia (PNFA) vs. CN (**b**) and semantic variant (SV) vs. CN (**c**). Associated structural connectivity reduction (FWE-corrected p-value < 0.001) is shown in panels (**d–f**) for the ipsilateral left (upper panel) and right (lower panel) hemisphere, where frontal regions are shown in red, the insula in

*Figure 2 continued on next page*

*Figure 2 continued*

light blue, the temporal lobe in yellow, subcortical regions in green, and parietal regions in dark blue. The line thickness corresponds to the statistical strength of the effect. Red = left-right, green = front back, blue = top down.

## GM atrophy in FTD variants

Significant differences, after correction for age, sex, ICV, and multiple comparisons, were observed for GM volume between CN and FTD variants (*Figure 3a–c*). BV (*Figure 3a*) had a widespread reduction in bilateral GM volume with the strongest effect seen in the insula, orbitofrontal, anterior cingulate, and prefrontal cortex (middle and inferior) while PNFA (*Figure 3b*) had a left predominant atrophy in the premotor part of the frontal cortex, the insula and prefrontal cortex (middle and inferior), and SV (*Figure 3c*) had a bilateral (but left predominant) atrophy of the whole temporal lobes and to a lesser extend insula atrophy. Taken together (*Figure 3d*), the three variants share overlapping GM atrophy in the insula, while BV and SV share atrophy in the temporal lobe and orbitofrontal cortex, and PNFA and BV share atrophy in the left prefrontal cortex (middle and inferior).

## Relationship between GM atrophy and WM microstructural impairment

The peak of the maximum GM atrophy for each variant was used as a seed (*Figure 4a–c* left panels) to investigate the relationship between GM and whole brain WM FC and FD across all participants. Streamlines associated to significant fixels after correction for multiple comparison are shown for the relationship with FC (middle panels) and FD (right panels). Independently of the seed location, a strong relationship was found between GM atrophy and reduced FC and FD for the inferior

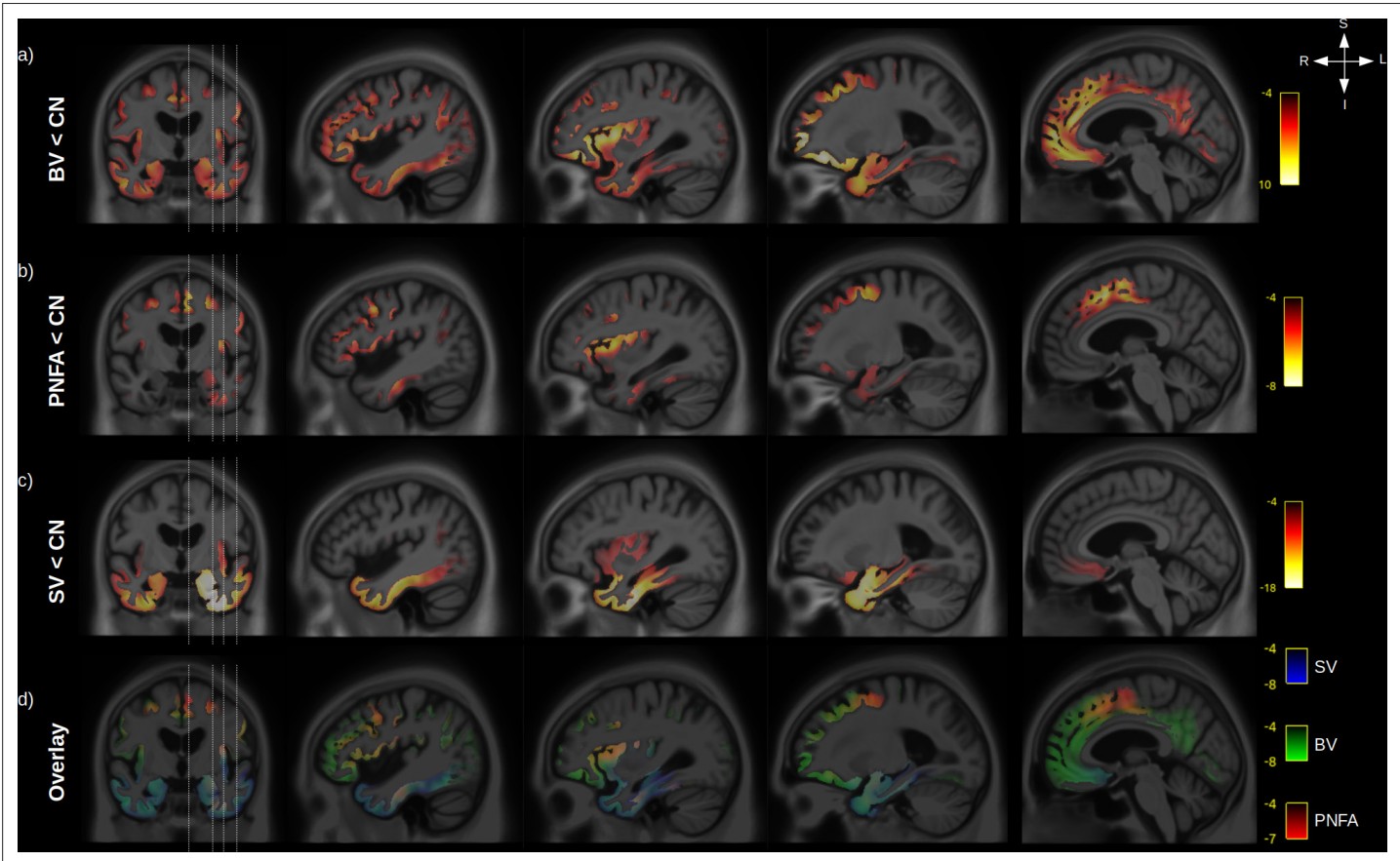

**Figure 3.** Grey-matter (GM) atrophy in fronto-temporal dementia (FTD) variants. Significant (RFT p-value < 0.05) GM volume decrease is shown for behavioural variants (BV) vs. normal elderly control (CN) (**a**), progressive non-fluent aphasia (PNFA) vs. CN (**b**) and semantic variant (SV) vs. CN (**c**). Legend are showing the magnitude of the voxelwise T values. An overlay of the statistical maps (**d**) is shown for BV (green), SV (blue), and PNFA (red), with associated T values colour bars.

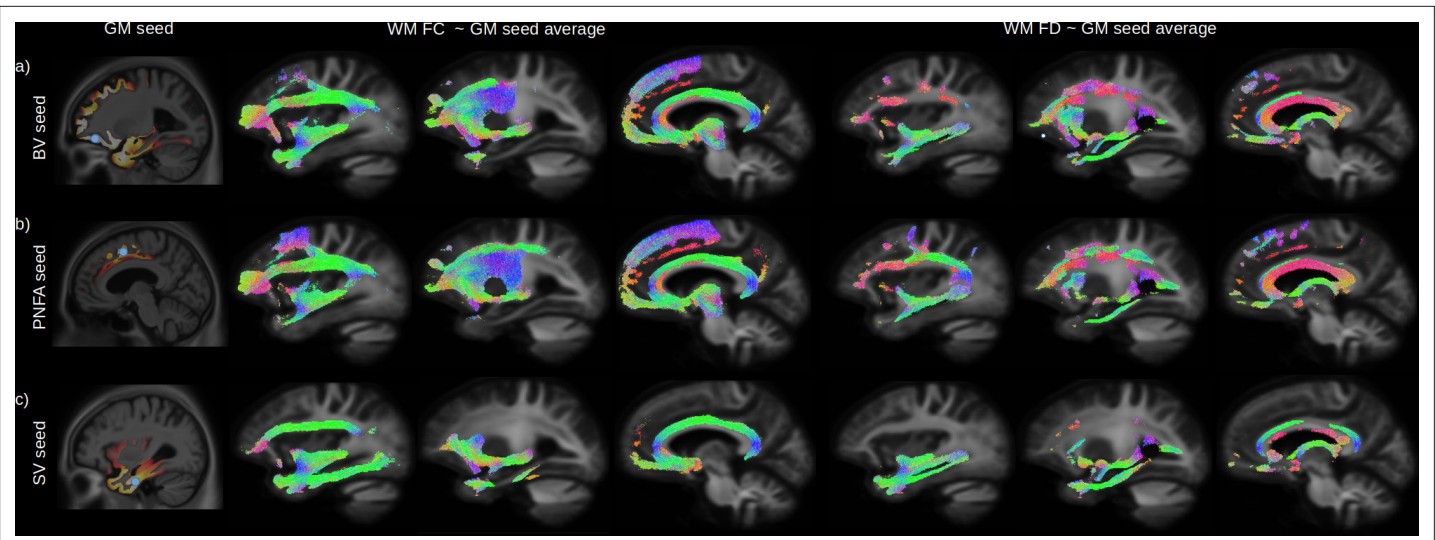

**Figure 4.** Relationship between grey-matter (GM) atrophy and white-matter (WM) microstructural impairment. The peak of the maximum GM atrophy for each variant (behavioural variant [BV], progressive non-fluent aphasia [PNFA], and semantic variant [SV]) was used as a seed (**a-c** left panels, blue dot) to investigate the relationship between GM and whole brain WM fibre cross section (FC) and fibre density (FD) across all participants. Streamlines associated with significant relationships (FWE-corrected p-value < 0.05) between the average GM volume of each seeds are shown for FC (middle panels) and FD (right panels). Streamlines are colour coded by direction.

fronto-occipital fasciculus, uncinate fasciculus, and superior longitudinal fasciculus. GM volume for the BV orbitofrontal seed (*Figure 4a*) was also associated with the inferior part of the precentral cortex, PNFA premotor seed (*Figure 4b*) with the precentral WM fibres, and SV inferior temporal seed with impairment in the inferior longitudinal fasciculus.

## Cognitive domains and WM microstructure

As highlighted earlier, our three factor model failed to reject the null hypothesis suggesting that this model provides a satisfactory explanation for the variation in this data. Therefore, in order to assess the relationship between impaired cognition and WM FD and FC, in the first step we performed a three common factor analysis across eight cognitive scores in patients only to represent these scores by three principal factors (*Figure 5a*). According to the loadings of this analysis, the first factor (Factor 1) was mainly related to semantic processing and comprised naming, category fluency, and verbal learning scores (delayed and immediate recall). The second factor (Factor 2) was mainly related to executive processing and comprised digit span, trail making, and letter fluency scores. The third factor (Factor 3) was only related to verbal fluency (category and letter). Although SV patients had on average a lower semantic factor score compared to BV (p = 0.034) and PNFA (p < 0.001) and a higher executive factor score (p < 0.001; vs. BV and PNFA), a large within-group variability can be noted for all factors (*Figure 5b*). Patients factor scores were used to investigate the relationship between each cognitive domain and structural connectivity as well as WM metrics FD and FC. After correction for age, sex, ICV, and multiple comparisons, the first (semantic) factor was significantly associated to FC in the uncinate fasciculus, the inferior fronto-occipital fasciculus, and the inferior longitudinal fasciculus (*Figure 5d*; upper panel). The semantic factor was also associated to reduced connectivity between the left GM temporal regions amongst themselves but also with the supramarginal, lateral orbitofrontal gyrus, and with the thalamus (*Figure 5c*). The second (executive) factor was significantly associated with a reduced FC in the superior longitudinal fasciculus, superior corona radiata, body of the corpus callosum, inferior frontal and precentral WM, and in fibres corresponding to the aslant tract (*Figure 5d*; lower panel). Reduced structural connectivity was predominantly observed between the left superior frontal gyrus and other GM frontal regions (pars orbitalis, pars triangularis, lateral orbitofrontal, rostral middle frontal, and precentral gyrus), accompanied by a reduced connectivity between left superior frontal gyrus and other cortices (insula, the superior temporal gyrus, and between the inferior parietal cortex) (*Figure 5c*). Although not shown in the figure, FD yielded similar spatial

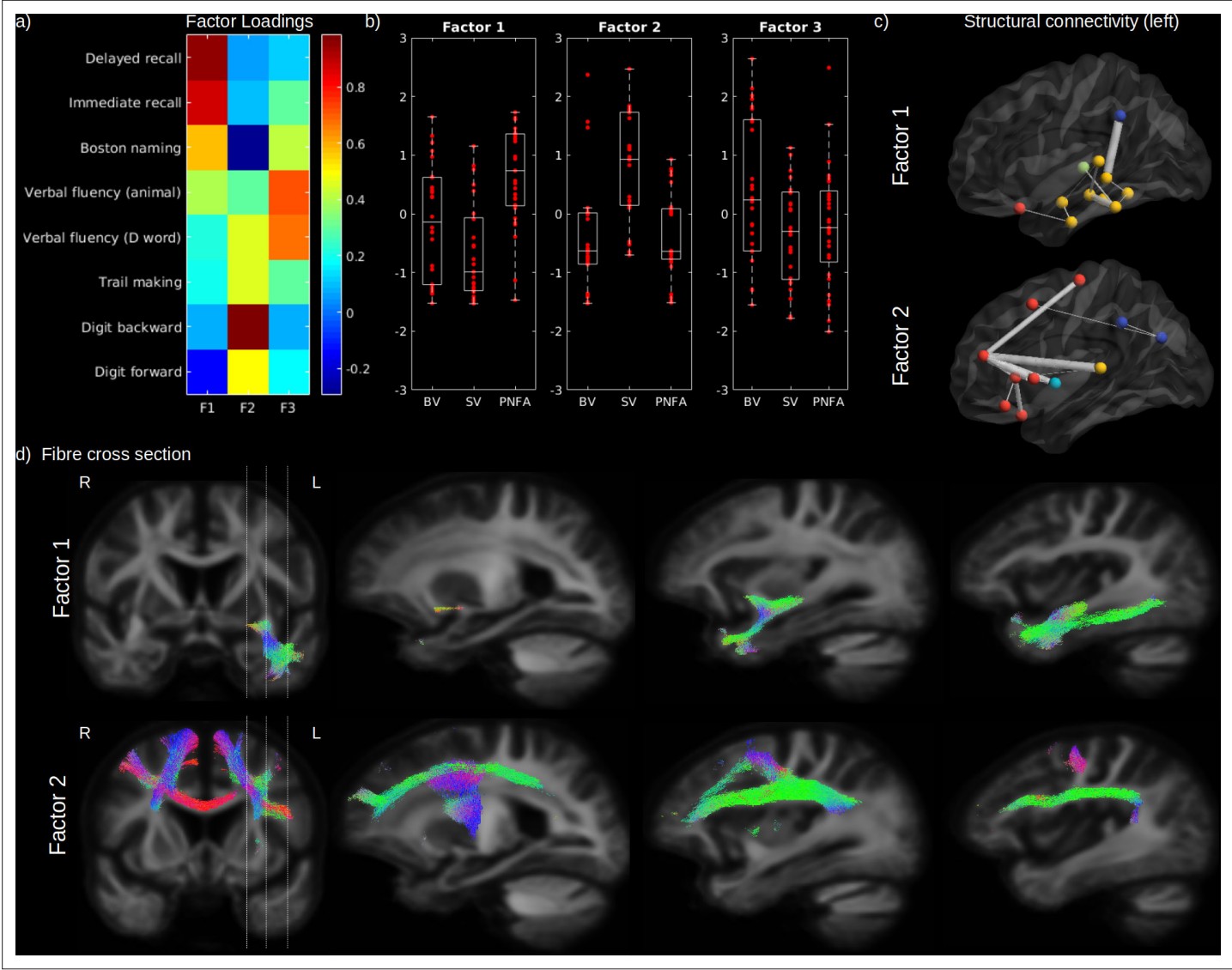

**Figure 5.** Cognitive domains and white-matter (WM) microstructure. The factor loadings for the common factor analysis of selected cognitive tests are shown in (**a**). Associated factors scores are shown in (**b**) for behavioural variant (BV), semantic variant (SV), and progressive non-fluent aphasia (PNFA), for Factor 1 (semantic processing; left panel), Factor 2 (executive processing; middle panel), and Factor 3 (verbal fluency; right panel). Significantly reduced structural connectivity (across all patients; FWE-corrected p-value < 0.01) is shown in (**c**) for the Factor 1 (upper panel) and Factor 2 (lower panel), for the ipsilateral left (upper panel) hemisphere connectivity, where frontal regions are shown in red, the insula in light blue, the temporal lobe in yellow, subcortical regions in green and parietal regions in dark blue. The line thickness corresponds to the statistical strength of the effect. Significant relationship (FWE-corrected p-value < 0.05) between the cognitive factors and fibre cross-section (FC) is shown in (**d**) with associated streamlines (colour coded by direction) for Factor 1 (upper panel) and Factor 2 (lower panel).

relationships than FC. No significant relationship could be found between the third factor (verbal fluency) and FC, FD, or structural connectivity (not shown).

## Relative contribution of GM and WM to predict cognitive impairment

In order to disentangle the contribution of WM and GM abnormalities on cognition impairment, we used the connectivity-based pair of GM regions previously associated to identified cognitive factors (Factors 1 and 2; FEW-p <0.001), where VBM-derived GM volume was averaged for the pair of regions and FD and FC averages were calculated from the fixel associated to the connecting streamlines (see Tract of interest analysis section). Example of pair of GM regions and fixel mask are shown in *Figure 6a–b* for the Factors 1 (semantic) and 2 (executive) respectively. After correcting for age, sex, and ICV, GM volume and FC better explained the variance for the first cognitive factor (39.2% and

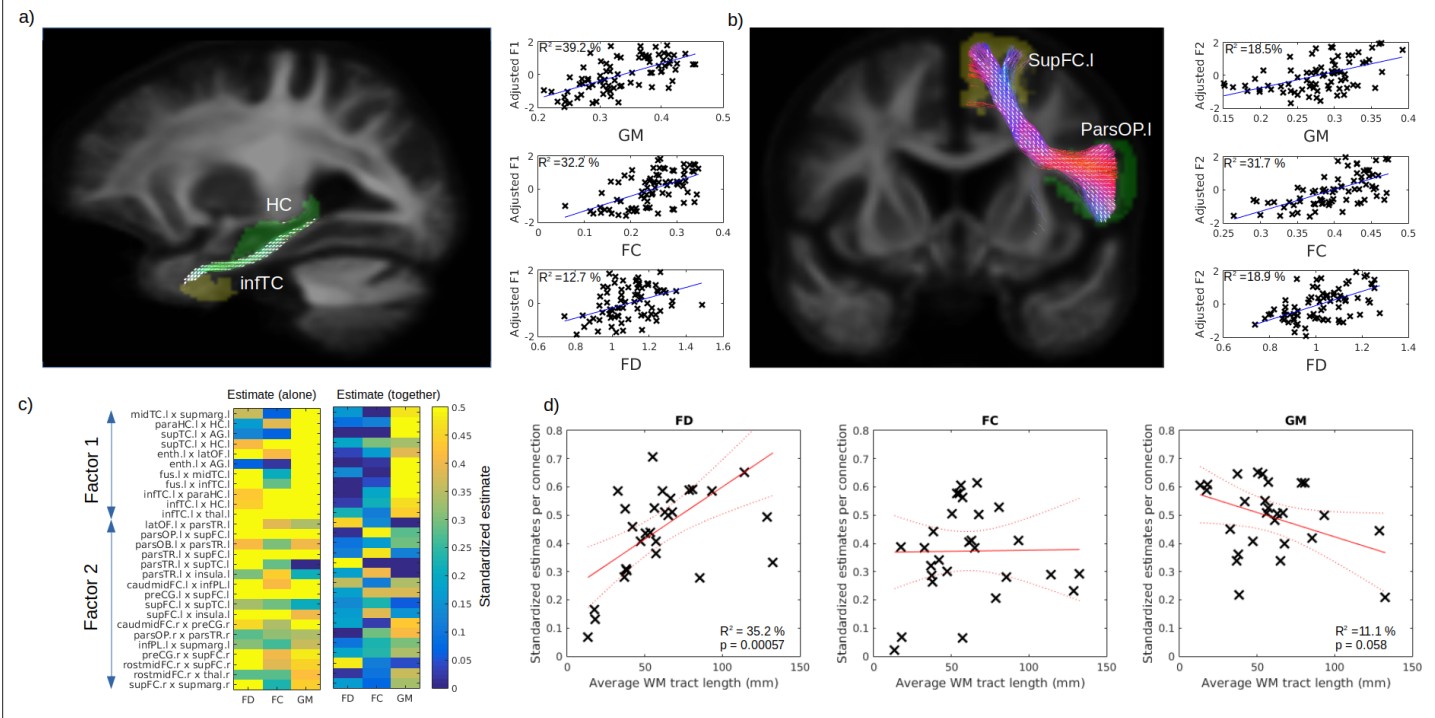

**Figure 6.** Respective contribution of grey matter (GM) and white matter (WM) to predict cognitive impairment. Example tract of interest, selected from the significant relationship between structural connectivity and cognitive factor, are shown in (**a–b**), for Factors 1 and 2, respectively. The pair of connecting cortical regions (green and yellow) and the fixel binary mask extracted from the streamlines connecting the pair of GM regions (white) are used to calculate the connection-specific relationship across subjects between the cognitive factors and the average regional GM volume (upper panel) and the average fibre cross-section (FC) and fibre density (FD) in the fixel binary mask (middle and lower panels, respectively). The standardized estimate of the relationship between the cognitive factors 1 and 2 is shown in (**c**) for all selected connections, where the left panel shows the values when FD, FC, and GM were used as single predictors and the right panels show the partial estimates when FD, FC, and GM were all included together in the model. The relationship between each connection estimate and their associated average fibre length is shown in (**d**) for FD (left panel), FC (middle panel), and GM (right panel).

32.2%, respectively) than FD (12.7%) (**Figure 6a**; right panels). On the other hand, for the second factor, FC explains the variance better (31.7 %) than GM (18.5%) and FD (18.9%) (**Figure 6b**; right panels). The comparisons of all the selected connection standardized estimates revealed that, for most connections, GM, FD, and FC could predict the cognition (**Figure 6c**; left panel). Importantly, when included together in the same general linear models (GLM) (**Figure 6c**; right panel), the contribution of the WM was reduced but not for all connections. We then tested the relationship between the standardized estimates and the average fibre length and found a positive relationship for FD (**Figure 6d**; left panel, p = 0.006) and a trend level negative relationship for GM (**Figure 6d**; right panel, p = 0.058), but not for FC (**Figure 6d** middle panel, p = 0.94).

## Discussion

In this study we aimed to quantify the relative contribution of WM and GM abnormalities as determinants of cognitive impairment in FTD clinical spectrum. We found that, although regional differences in WM properties were observed in all variants, all FTD cases had FD or FC abnormalities across a large WM network connecting the frontal and temporal cortices. Furthermore, these WM abnormalities were linked to patterns of GM atrophy and cognitive decline across FTD variants. The differential contributions of WM and GM on cognition depended on the length of WM fibre. Although both GM and WM abnormalities contribute to FTD symptoms, these results highlight the importance of WM FD and FC in FTD pathophysiology.

## FD and FC and structural connectivity phenotypes in variants of FTD

Our study identified a large network (uncinate fasciculus, superior longitudinal fasciculus, inferior fronto-occipital fasciculus, cingulum, and corpus callosum) of WM impairment being shared across FTD variants, extending findings from previous reports (*Agosta et al., 2015*). Also in line with previous literature, we found SV to have additional impairments in the inferior fronto-occipital fasciculus (*Acosta-Cabronero et al., 2011*; *Agosta et al., 2010*; *Galantucci et al., 2011*; *Matsuo et al., 2008*; *Whitwell et al., 2010*), BV in the frontal cortex (*Agosta et al., 2012*; *Mahoney et al., 2014*; *Piguet et al., 2011*; *Yu et al., 2019*; *Zhang et al., 2009*), and PNFA in SMA WM fibres (*Agosta et al., 2015*; *Mahoney et al., 2013*). Our structural connectivity results were in agreement with the fixel-based results suggesting that both techniques were able to detect WM impairments in FTD. In addition, we found that one of the largest reductions in structural connectivity was between thalamo-frontal regions, supporting the finding that thalamic atrophy is a prominent feature of FTD (*Diehl-Schmid et al., 2019*) and that it is common across episodic and genetic mutation (*Bocchetta et al., 2018*).

## Cognitive factor across variants

Across the FTD spectrum, we found that a common semantic factor explained the variance of scores in the immediate and delayed verbal memory test, picture naming, and categorical verbal fluency. This is in line with several studies showing poorer lexical retrieval of semantically degraded words vs. semantically intact words (*Jefferies et al., 2004*; *Knott et al., 2000*; *Patterson et al., 1994*), suggesting that semantic information contributes significantly in (phonological) lexical activation. The second cognitive factor (executive) explained the variance of scores in the modified trail making processing time, digit span (forward and backward), and phonemic, but not categorical, verbal fluency. TMT measures multiple executive functions, including attention, processing speed, set-shifting, and digit span (forward and backward), and is typically used as an attentional/working memory measure, while phonemic verbal fluency contains both a working memory/executive and a language component. A third factor grouped the two verbal fluency tests together, however this factor only partly explains the variance of each test as they also weighed on semantic processing (for category fluency) and executive functioning (letter fluency), supporting the dual nature of the verbal fluency test (*Whiteside et al., 2016*) even in non-demented individuals.

## WM and cognition in FTD

### Semantic processing

In the present study, we found evidence of the relationship between semantic deficits and WM impairment in the left uncinate fasciculus, inferior longitudinal fasciculus, and inferior fronto-occipital fasciculus, across all variants. The uncinate fasciculus (connecting the orbitofrontal cortex to the temporal pole) has been associated with semantic processing in many studies; see *Papagno, 2011*, for a review. Brain stimulation studies (*Duffau et al., 2008*; *Duffau et al., 2005*) and post-mortem fibre dissection studies *Martino et al., 2010* have linked the ventral subcomponent of the inferior fronto-occipital fasciculus (connecting the frontal lobe to occipital associative extrastriate cortex and the temporo-basal region) and semantic processing. Prior studies led to inconsistent results regarding the involvement of the inferior longitudinal fasciculus, connecting the ventro-anterior temporal lobes, to several occipital regions (fusiform gyrus, lingual gyrus, and dorsolateral occipital cortex); see *Cocquyt et al., 2020*, for a recent review. Our structural connectivity analysis revealed that the connectivity between the left inferior temporal cortex and the thalamus may also be involved in semantic processing. In general, our findings support the so-called hub model for the semantic processing where the anterior temporal pole represents a unique trans-modal hub receiving and assembling information from different modality specificity brain regions via specific WM connections (*Patterson et al., 2007*). Interestingly, it was also proposed (*Ralph et al., 2017*) that graded deficit in semantic processing is dependent on the WM fasciculi connecting the anterior temporal lobe to the cortex, where the uncinate fasciculus, superior longitudinal fasciculus, and inferior fronto-occipital fasciculus would convey either social, verbal, or visual semantic processing, respectively. Altogether, our findings support the hub hypothesis and suggest that it could be expanded further by considering subcortical contribution to the model.

## Executive processing

Executive processing is a prominent frontal function and not surprisingly it is severely affected in the FTD clinical spectrum. We found that the executive function impairment was associated with disruption of WM tracts in the frontal lobe, specifically in superior longitudinal fasciculus, superior corona radiata, the body of the corpus callosum, inferior frontal and supplementary motor WM, and the aslant tract. The superior longitudinal fasciculus (connecting the frontal lobe to temporal and parietal cortices) has previously been associated to processing speed (*Turken et al., 2008*) and working memory (*Rizio and Diaz, 2016*) and impairment of the corona radiata (connecting the prefrontal cortex to the basal ganglia and thalamus) has also been associated to executive dysfunction (*Hua et al., 2014*; *Moeller et al., 2015*). Interestingly, the aslant tract (connecting the SMA with the inferior frontal cortex) has been associated with the self-initiated movement and speech production (*Kinoshita et al., 2015*) and its integrity correlated with the amount of distortion errors that PNFA patients made in spontaneous speech (*Mandelli et al., 2014*). Moreover, our results suggest that the contribution of WM to executive deficits increases with the length of these WM tracts. Reduced WM integrity in large-scale WM tracts was the major player of executive dysfunction in the FTD population. Interestingly, large-scale WM tracts are also particularly vulnerable to WM vascular disease, as observed in post-mortem studies (*O'Brien et al., 2002*). Moreover, chronic ischemic microvascular lesions, depicted as diffuse WM hyperintensities in brain MRI scans, are independently associated with impairment of executive function (*Young et al., 2008*). Our results could thus suggest that patients with FTD and compromised large-scale WM fibres might be particularly vulnerable to additional vascular pathology. Thus, our findings highlight the importance for controlling vascular risk factors in FTD patients in order not to potentiate the underlining executive dysfunction. Alternatively, given that the majority of FTD patients are younger and less likely to have significant vascular disease, it is possible that the tract is more vulnerable to a degree of degenerative pathology.

## Relationship between GM, WM, and cognition

As WM and GM impairment are too often considered in isolation, one of the goals of our study was to investigate their relative contribution to neurodegeneration. We found that the magnitude of GM atrophy was strongly related to the impairment of WM networks. This was also observed in both AD and FTD using canonical correlation analysis (*Avants et al., 2010*). Modeling the combined contribution of GM and WM to cognition is not straightforward because of the lack of spatial overlap between these modalities. To overcome this challenge, we took advantage of a common connectivity space that encompasses both structural connectivity and fibre-specific WM pathways. This construct allowed us to select anatomically relevant connections, to extract their average regional GM volume and streamlines-based respective FC and FD for predicting their respective contribution on the cognitive domain. Interestingly, within short connections, the contribution of GM atrophy was dominant, while WM FD gained in importance as a function of fibre length. This finding supports a framework in which cognitive functions involving short-range circuits are mostly affected by local GM atrophy, while cognitive processes mediated by long-range fibres are more vulnerable to WM impairment. Thus, our results support the critical importance of considering both GM and WM alterations for a better understanding of distinctively spatially distributed cognitive alterations in neurodegeneration.

## FBA applied to FTD

To assess WM FD and FC, we used a novel fixel-based approach where individual fibre populations, even within the same voxel, can be assessed independently. Older diffusion tensor imaging (DTI) techniques, although historically invaluable in offering the earliest opportunities to non-invasively investigate some microstructural properties of WM and their alteration in aging and disease, suffered from the inability to resolve crossing fibres. It was shown that traditional DTI may lead to artefactual findings in neurodegenerative disorders (*Mito et al., 2018*; *Tournier et al., 2008*), both false positive and false negative. This severely limits the extent to which such DTI findings can be interpreted or even safely relied upon. Novel techniques, such as constrained spherical deconvolution (*Tournier et al., 2007*; *Tournier et al., 2004*) and FBA *Raffelt et al., 2015* have greatly improved the accuracy of dMRI processing and whole brain statistical analysis. The associated metrics, FD and FC, were recently proposed to capture different properties of the WM fibre (*Raffelt et al., 2017*). FD is considered a measure of WM microstructure, while FC is related to macroscopic fibre bundle morphometric

change. Although these measures are typically not independent, they can provide insight on different types of WM impairment and have successfully been applied to Alzheimer's disease (*Mito et al., 2018*). In the context of FTD, we found that both FD and FC were reduced in similar WM regions, which suggest that both fibre atrophy and axonal depletion that are part of the disease.

## Strengths and limitations

This study has several strengths. To our knowledge, this is the first study applying an FBA to analyse WM impairments across FTD phenotypes, thus broadening the biological interpretation of WM alterations in the pathophysiology of this disease. There is a growing body of evidence describing WM degeneration in several cortical diseases. However, most studies did not investigate the relationship between specific cognitive domains, whole brain WM properties, and structural connectivity. Therefore, our study provided a more complete picture of specific WM tracts involved in core FTD cognitive impairment. Finally, the use of an innovative connection-based framework, allowing for the quantification of the simultaneous contribution of WM and GM abnormalities on cognitive deficits in FTD, also expanded the knowledge about multimodal contribution to cognition. The main limitations are due to a limited number of subjects and the lack of longitudinal data. Although patients were clinically assessed with the highest standards, the lack of genetic or pathological information precludes any association between the proteins involved in the etiology of FTD, such as tau and TDP-43, and WM fibres. Furthermore, as the data obtained from Frontotemporal Lobar Degeneration Neuroimaging Initiative (FTLDNI) are the result of a multicentric collaboration, differences in scanners, protocols, and center-specific differences could impact our findings. Nonetheless, before the release of the data, a quality control was conducted. In addition, while the number of diffusion gradient directions (60) and the b-value (2000) are suitable to obtain a good overall quality of the WM FODs, the spatial resolution was limited to 2.2 mm isotropic voxels. Since some bundles of white fibres are only a few mm wide, significant group differences in these bundles are difficult to detect at the resolution of the data used in the present study. This, however, was the maximum resolution that could be obtained for this signal while still maintaining a good signal-to-noise ratio. Finally, although our imaging analyses controlled for age, sex, and ICV but not for clinically relevant variables including disease duration and symptom severity, as these would be artificial and could potentially bias the results of a study with such a diverse clinical population, this imposes a limitation on the interpretation of the results presented in this study.

## Conclusion

In conclusion, our results support the importance of WM tract disruption to the core cognitive symptoms associated with FTD. While semantic symptoms were mainly dependent on short-range WM fibre disruption, long-range WM fibres damage was the major contributor to executive dysfunction. As large-scale WM tracts, which are particularly vulnerable to vascular disease, were highly associated with executive dysfunction, our findings highlight the importance of controlling for risk factors associated with deep WM disease, such as vascular risk factors, in patients with FTD in order not to potentiate underlying executive dysfunction.

**Table 1.** Demographics.

|  | CN (N = 68) | BV (N = 28) | SV (N = 26) | PFNA (N = 30) |
|---|---|---|---|---|
| Age (year) | 61.8 (8.2) | 60.6 (6.4) | 62.6 (6.0) | 68.3 (7.4) |
| Sex (female) | 60.3 % | 21.4 % | 42.3 % | 63.3 % |
| CDR language | – | 0.84 (0.53) | 1.04 (0.47) | 1.38 (0.66) |
| CDR behaviour | – | 1.48 (0.72) | 0.98 (0.48) | 0.41 (0.46) |
| CDR sum of boxes | – | 5.96 (2.78) | 3.54 (2.02) | 1.59 (1.55) |
| MMSE | 29.2 (0.8) | 24.3 (3.7) | 25.8 (3.6) | 25.3 (4.9) |

## Materials and methods

### Study sample

All data were obtained from the FTLDNI, through the LONI portal (http://adni.loni.usc.edu). FTLDNI is a multicentric longitudinal database, collecting MRIs, PET, and CSF biomarkers in FTD patients and age-matched controls. All patients were clinically diagnosed by a multidisciplinary consensus panel (*Ljubenkov et al., 2018*; *Staffaroni et al., 2019*). For the present analysis, we included a total of 155 participants with cross-sectional DWI sequence passing quality control. The dataset comprised 68 normal elderly control (CN), 28 BV, 30 PNFA, and 26 SV FTD patients (see *Table 1* for demographics).

### MRI acquisition

A total of 65 volumes (diffusion-weighted images for 60 gradient directions at b = 2000 s/mm$^2$ and 5 images at b = 0 s/mm$^2$) were acquired on a Siemens Trio Tim with the following parameters: repetition time/echo time = 6600/86 ms, 2.2 mm isotropic voxels, phase encoding direction = AP. A 3D MPRAGE image (1 mm isotropic voxels, repetition time/echo time = 2300/2.98 ms, and flip angle = 9 degrees) was also used to measure GM volume.

### dMRI processing

We implemented preprocessing and analysis steps of a state-of-the-art FBA pipeline (*Dhollander et al., 2021*). All dMRI data were preprocessed using MRtrix3 (*Tournier et al., 2019*). Preprocessing steps included denoising (*Veraart et al., 2016*), Gibbs ringing correction (*Kellner et al., 2016*), eddy-current and motion correction (*Andersson and Sotiropoulos, 2016*), and bias field correction (*Tustison et al., 2010*). Response functions for single-fibre WM as well as GM and CSF were estimated from the data themselves using an unsupervised method (*Dhollander et al., 2019*). Single-shell 3-tissue CSD was performed to obtain WM-like FODs as well as GM-like and CSF-like compartments in all voxels (*Dhollander and Connelly, 2016*), using MRtrix3Tissue (https://3Tissue.github.io), a fork of MRtrix3 (*Tournier et al., 2019*). The resulting WM-like FOD, GM-like and CSF-like images were used to perform multi-tissue informed log-domain intensity normalization (*Figure 1a*). A cubic b-spline interpolation was used to upsample the WM FOD images to 1.3 mm isotropic voxels. A study-specific template was created using the WM FOD images from 30 NC to which all subjects' FOD images were subsequently non-linearly registered (*Raffelt et al., 2012a*; *Raffelt et al., 2011*). Finally, the WM FOD template was used to generate a whole brain probabilistic tractogram (*Tournier et al., 2010*) which was then filtered from 20 million tracts to 2 million tracts to reduce reconstruction bias (*Smith et al., 2013*).

### Fixel-based metrics

We used the FBA framework (*Raffelt et al., 2017*; *Raffelt et al., 2012b*) to compute the FD and the FC at the fixel level (*Figure 1b*). A 'fixel' here refers to a 'fibre population in a voxel'; hence, when multiple fibres are crossing in the same voxel, they each still have individual measures of FD and FC. Interestingly these metrics provide complementary information about the WM. Namely, FD-based differences can be interpreted as intra-axonal microstructural alterations, while FC-based differences can be attributed to macroscopic changes of a fibre bundle, that is, a tract that is atrophied or hypertrophied in respect to the WM FOD template.

### Structural connectivity analyses

A probabilistic tractography algorithm (*Tournier et al., 2010*) with dynamic seeding (*Smith et al., 2015*) was used to generate 20 million tracks for each participant's WM FODs in the template space (*Figure 1c*). The tractogram was subsequently filtered using SIFT (*Smith et al., 2013*) until the algorithm reaches convergence. We used the Desikan-Killiany (DKT) GM atlas to compute the amount of fibres connecting 68 GM regions (*Desikan et al., 2006*). An affine transformation was first calculated from the MNI ICBM152 WM parcellation to the diffusion template space (*Figure 1e*). The affine transform was applied to the DKT atlas to bring it in diffusion template space (*Figure 1f*) and the atlas was corrected by the amplitude of the template WM FOD, where amplitudes higher than 0.1 were set to zero. A visual inspection of the resulting GM atlas insured that all GM regions were well represented. Structural connectomes were calculated as the total number of fibres paths connecting each pair of

GM regions (*Figure 1g*). The results of the statistical analysis performed on the connectomes were visualized using BrainNet Viewer (*Xia et al., 2013*).

## GM voxel-based morphometry

T1 anatomical images were segmented in GM, WM, and CSF tissue probability images using the SPM12 segmentation tool (https://www.fil.ion.ucl.ac.uk/spm/doc/biblio/). A study-specific brain template was then calculated using the GM and WM probabilities from 30 CN using the Dartel toolbox (*Ashburner, 2007*). Each individual GM map was non-linearly registered to the CN template (*Figure 1d*). GM probabilities were modulated and filtered using a full width half maximum of 8 mm. ICV was defined as the sum of GM, WH, and CSF probabilities images in native T1 space.

## Cognitive tests

In order to clinically characterize the FTD patients, the following cognitive scores were used: the total correct immediate (30 s) and delayed (10 min) items recall of the California Verbal Learning Test, the total Boston naming correct score, the semantic verbal fluency (animal), the phonemic verbal fluency (d words), the modified trail making completion time, the forward and backward digit span. A maximum likelihood common factor analysis ('factoran' function in Matlab, with varimax rotation) was used to obtain a parsimonious representation of all available cognitive scores, as we wanted to obtain an explanatory model for the correlations amongst these scores. A two common factor hypothesis was first rejected (approximate chi-squared test; $p < 0.05$) while a three-factor model fails to reject the null hypothesis (approximate chi-squared test; $p < 0.42$), suggesting that the latter model provides a satisfactory explanation of the covariation in these data (see *Figure 5a* for the factor loadings results). The factor scores were calculated using a weighted least score estimate.

## Tract of interest analysis

Using a matrix of regions, pairs of GM regions were made, based on the significance of their connectivity. These selected connectivity-based pairs of GM regions were used to extract the tracts connecting them, which allows to investigate the tract-specific relations between GM volume, FD, and FC and cognition. Using the template filtered tractogram (2 M streamlines), we extracted the streamlines assigned to the pair of selected GM regions (*Figure 1i*). The resulting streamlines were then automatically thresholded into a binary fixel mask using an automated optimal threshold (*Ridgway et al., 2009*). The connectivity-based FC and FD values were then averaged in the mask. The GM volume was assessed by calculating the average GM VBM values of the connecting regions (see *Figure 6a–b* for a graphical representation). To investigate all the selected connections as a whole, we standardized GM, FC, and FD across all connections and used repeated GLM to obtain the prediction estimate for their respective cognitive factors adjusted for age, sex, and ICV (*Figure 1i*). Finally, we calculated the average streamline length for each tract, which best represents the overall length of the tract.

## Statistical analysis

Fixel-wise whole brain characterization of the relationship between FC, FD, diagnosis, and cognition was carried out using the connectivity-based fixel enhancement method (*Raffelt et al., 2015*). For the structural connectivity analysis, a common connectivity mask was generated for the top 20% connections of the population template. Relationship between the connectivity, diagnosis, and cognition was calculated using the network-based statistical enhancement method (*Vinokur et al., 2015*). For both methods, family-wise-corrected p-values were obtained via permutation testing (n = 1000). VBM analyses were performed using VoxelStats (*Mathotaarachchi et al., 2016*). Correction for multiple comparisons was performed using random field theory with a cluster threshold (after correction) of $p < 0.01$. Analyses of cognition were performed in patient groups only. All statistical models were corrected for age, sex, and ICV.

# Additional information

## Funding

| Funder | Grant reference number | Author |
| --- | --- | --- |
| National Institutes of Health | R01 AG032306 | Pedro Rosa-Neto |

The funders had no role in study design, data collection and interpretation, or the decision to submit the work for publication.

## Author contributions

Melissa Savard, Conceptualization, Formal analysis, Investigation, Methodology, Validation, Visualization, Writing - original draft; Tharick A Pascoal, Stijn Servaes, Thijs Dhollander, Yasser Iturria-Medina, Min Su Kang, Paolo Vitali, Joseph Therriault, Sulantha Mathotaarachchi, Andrea Lessa Benedet, Serge Gauthier, Writing – review and editing; Pedro Rosa-Neto, Conceptualization, Funding acquisition, Investigation, Methodology, Project administration, Supervision, Writing – review and editing

## Author ORCIDs

Stijn Servaes http://orcid.org/0000-0002-4431-957X
Thijs Dhollander http://orcid.org/0000-0003-3088-3636
Yasser Iturria-Medina http://orcid.org/0000-0002-9345-0347
Sulantha Mathotaarachchi http://orcid.org/0000-0001-9391-4503
Pedro Rosa-Neto http://orcid.org/0000-0001-9116-1376

## Ethics

Human subjects: All data were obtained from the Frontotemporal Lobar Degeneration Neuroimaging Initiative (FTLDNI), through the LONI portal (http://adni.loni.usc.edu). FTLDNI is a multicentric longitudinal database, collecting MRIs, PET and CSF biomarkers in FTD patients and age-matched controls. The investigators at NIFD/FTLDNI contributed to the design and implementation of FTLDNI and/or provided data, but did not participate in the analysis or writing of this report.

## Decision letter and Author response

Decision letter https://doi.org/10.7554/eLife.73510.sa1
Author response https://doi.org/10.7554/eLife.73510.sa2

# Additional files

## Supplementary files

• Transparent reporting form

## Data availability

All data were obtained from the Frontotemporal Lobar Degeneration Neuroimaging Initiative (FTLDNI) and are available through the LONI portal (http://adni.loni.usc.edu). FTLDNI is a multicentric longitudinal database, collecting MRIs, PET and CSF biomarkers in FTD patients and age-matched controls.

The following previously published dataset was used:

| Author(s) | Year | Dataset title | Dataset URL | Database and Identifier |
| --- | --- | --- | --- | --- |
| Howard R | 2010 | FTLDNI | http://4rtni-ftldni.ini.usc.edu/ | ftldni, 4rtni |

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
