## [Editor Report]

This study explores how the pathophysiology of frontotemporal dementia, a collection of younger-onset dementias, impacts grey and white mater brain integrity, and how such changes relate to discrete aspects of cognition. The authors used whole-brain fixed-based analysis, structural connectivity analysis of white matter tracts, alongside voxel-based morphometry of grey matter atrophy. Overall, semantic impairment was found to associate with relatively short-range white matter dysfunction, while executive dysfunction was related to long-range white matter fibres.

---

## [Decision Letter]

**Decision letter after peer review:**

Thank you for submitting your article "Impact of long- and short-range fiber depletion on the cognitive deficits of fronto-temporal dementia" for consideration by *eLife*. Your article has been reviewed by 3 peer reviewers, one of whom is a member of our Board of Reviewing Editors, and the evaluation has been overseen by Jeannie Chin as the Senior Editor. The following individual involved in review of your submission has agreed to reveal their identity: Timothy Rittman (Reviewer #3).

Essential revisions:

1. The Introduction offers an incomplete picture of grey and white matter contributions to cognition in FTD. A number of studies have related both grey and white matter atrophy to discrete aspects of cognition in bvFTD and SD including disinhibition (Hornberger et al., 2011), moral reasoning (Strikwerda-Brown et al., 2021) and white matter changes over time (Lam et al., 2014) across the syndromes of interest here. A more balanced and comprehensive picture of previous work should be presented.

2. Participants: Table 1 provides a very limited overview of participant demographics and clinical characteristics. I would like to see much more information provided regarding these cases. For example, are the semantic dementia cases left- or right-predominant? What expressive language dysfunction tests were conducted to confirm the PNFA diagnosis? Was any cognitive testing conducted for Control participants? Are these groups matched for age, education, sex, etc? Were any tests of social cognition, semantic comprehension, or verbal repetition conducted? These are crucial for the differential diagnoses of the FTD subtypes.

3. Factor analysis: More information regarding the criteria used to extract the factors is required. Importantly, there is no clear separation between the 3 groups using these factors. This raises concerns regarding the suitability of the factors to take forward into the imaging, compounded by the fact that there is marked overlap in grey matter atrophy between the three patient groups. A clear demonstration of the specificity of the brain-behaviour associations either to a particular variant or cognitive domain is needed.

4. Imaging: As previously reported, FTD patients frequently show the presence of white matter lesions in variable locations and of different etiology, size, and severity. What is the impact of white matter lesions on the accuracy and precision of image registration and fiber estimation in this study? Did the authors assess the lesion load and/or use a lesion mask?

5. Imaging acquisition: The FTLDNI dataset includes DTI scans acquired with similar but not identical acquisition protocols (e.g., TR 6600 or 8200). Did the authors check if the 155 scans included in the study were all acquired using an identical acquisition protocol?

6. DTI: While the number of diffusion gradient directions (60) and the b-value (2000) are suitable to obtain a good overall quality of the WM FODs, I am concerned about the spatial resolution (2.2 mm isotropic voxels). Since some bundles of white fibers are only a few mm wide, significant group differences in these bundles are difficult to detect at the resolution of the data used in the present study. The authors should clarify this aspect and interpret the results carefully.

7. Head motion: Previous investigations have shown the impact of head motion on estimated diffusion values. Even with a rigorous quality check and eddy current correction, head motion can significantly influence diffusion values. Have the authors assessed and controlled for head motion?

8. On page: 9, it was not clear how the pairs of GM regions were defined/selected.

9. The imaging analyses controlled for age, sex and ICV but not for clinically relevant variables including disease duration, symptom severity, which limits the ability to interpret the results.

10. Discussion: Finally, the discussion of vulnerability of the aslant tract to vascular pathology contributing to FTD pathology seems unsupported. This seems unlikely given that the majority of FTD patients are younger and less likely to have significant vascular disease. Could it be that the tract is vulnerable to any form of degenerative pathology?

---

## [Author Response]

Essential revisions:1. The Introduction offers an incomplete picture of grey and white matter contributions to cognition in FTD. A number of studies have related both grey and white matter atrophy to discrete aspects of cognition in bvFTD and SD including disinhibition (Hornberger et al., 2011), moral reasoning (Strikwerda-Brown et al., 2021) and white matter changes over time (Lam et al., 2014) across the syndromes of interest here. A more balanced and comprehensive picture of previous work should be presented.

We thank the reviewer for this valuable contribution and elaborated on this by integrating these references into the introduction.

2. Participants: Table 1 provides a very limited overview of participant demographics and clinical characteristics. I would like to see much more information provided regarding these cases. For example, are the semantic dementia cases left- or right-predominant? What expressive language dysfunction tests were conducted to confirm the PNFA diagnosis? Was any cognitive testing conducted for Control participants? Are these groups matched for age, education, sex, etc? Were any tests of social cognition, semantic comprehension, or verbal repetition conducted? These are crucial for the differential diagnoses of the FTD subtypes.

We appreciate the reviewer’s comment regarding the participants’ demographic overview. We would like to highlight that this was done on purpose, considering the fact that this study does not focus on any specific type for FTD. An exhaustive description of these characteristics will detract from the main contributions of this paper as its goal is to search for commonalities across different phenotypes.

Furthermore, as the data was obtained from the Frontotemporal Lobar Degeneration Neuroimaging Initiative (FTLDNI) database, we are limited to including parameters that are present within this specific dataset. More information regarding this dataset can be found in Manera et al., 2021, Yu et al., 2021 Illán-Gala et al., 2019.

3. Factor analysis: More information regarding the criteria used to extract the factors is required. Importantly, there is no clear separation between the 3 groups using these factors. This raises concerns regarding the suitability of the factors to take forward into the imaging, compounded by the fact that there is marked overlap in grey matter atrophy between the three patient groups. A clear demonstration of the specificity of the brain-behaviour associations either to a particular variant or cognitive domain is needed.

We agree that this is indeed a crucial part of our manuscript. We therefore further clarified this as follows:

“As highlighted earlier, our 3 factor model failed to reject the null hypothesis suggesting that this model provides a satisfactory explanation for the variation in this data. Therefore, in order to assess the relationship between impaired cognition and WM fiber density (FD) and fiber cross-section (FC), in a first step we performed a 3 common factor analysis across eight cognitive scores in patients only to represent these scores by 3 principal factors (Figure 5a). According to the loadings of this analysis, the first factor (F1) was mainly related to semantic processing and comprised naming, category fluency and verbal learning scores (delayed and immediate recall). The second factor (F2) was mainly related to executive processing and comprised digit span, trail making and letter fluency scores. The third factor (F3) was only related to verbal fluency (category and letter). Although SV patients had on average a lower semantic factor score compared to BV (p=0.034) and PNFA (p < 0.001) and a higher executive factor score (p < 0.001; vs BV and PNFA), a large within group variability can be noted for all factors (Figure 5b). Patients factor scores were used to investigate the relationship between each cognitive domain and structural connectivity as well as WM metrics FD and FC.”

4. Imaging: As previously reported, FTD patients frequently show the presence of white matter lesions in variable locations and of different etiology, size, and severity. What is the impact of white matter lesions on the accuracy and precision of image registration and fiber estimation in this study? Did the authors assess the lesion load and/or use a lesion mask?

We would like to address these points brought forward by the reviewer in multiple parts. Firstly, the fiber orientation density (FOD) estimation can cope with the presence of WM lesions due to the 3-tissue spherical deconvolution approach: non-axonal diffusion signal contributions are captured in the other (non-WM) compartments of the model. Put differently, these non-axonal signals are thus “filtered out” of the WM FOD that is used for the subsequent image registration (and further analysis). Within lesions, we usually still see entirely intact WM FODs that are simply a bit smaller in size (but not distorted in “shape”), proportional to the remaining intact axons in these lesions.

Secondly, the image registration is directly guided by the WM FODs themselves. As the WM FODs in lesions still reflect the underlying fibre orientations via their undistorted shape, the image registration has no issues relying on these FODs to match them between subjects with and without lesions in any given WM area. This is quite common in fixel-based analyses, even in cohorts with much larger lesions (e.g. Alzheimer’s disease or even stroke; see Dhollander et al., 2021 for various examples).

Finally, when it comes to registration, as these lesions are less than 10cc, there is no noticeable effect in our registration quality (DeCarli et al., 1995).

5. Imaging acquisition: The FTLDNI dataset includes DTI scans acquired with similar but not identical acquisition protocols (e.g., TR 6600 or 8200). Did the authors check if the 155 scans included in the study were all acquired using an identical acquisition protocol?

We would like to thank the reviewer for noticing this. This is a multicentric study with an imaging core located at the Center for Imaging of Neurogenerative Diseases located at the Sandler Neuroscience Building on the Mission Bay Campus, San Francisco with a quality control that was conducted before the data release. Like any multicentric study there are differences due to different scanners, protocols and center specific differences. We therefore now acknowledge this limitation in our results:

“Furthermore, as the data obtained from FTLDNI are the result of a multicentric collaboration, differences in scanners, protocols and center specific differences could impact our findings. Nonetheless, before the release of the data, a quality control was conducted.”

6. DTI: While the number of diffusion gradient directions (60) and the b-value (2000) are suitable to obtain a good overall quality of the WM FODs, I am concerned about the spatial resolution (2.2 mm isotropic voxels). Since some bundles of white fibers are only a few mm wide, significant group differences in these bundles are difficult to detect at the resolution of the data used in the present study. The authors should clarify this aspect and interpret the results carefully.

We agree with the concerns of the reviewer. However, 2.5 mm was the max resolution that we could obtain for this signal to obtain a good STN ratio. We will therefore address the spatial resolution as one of the limitations of our study. We would like to note that this resolution is very common for diffusion MRI studies with higher b-values (in order to keep the signal-to-noise ratio under control). This can be seen in the fixel-based analysis review that was done by one of the co-authors of this manuscript (Dhollander et al., 2021) available here:

https://www.sciencedirect.com/science/article/pii/S1053811921006923.

In addition, while the number of diffusion gradient directions (60) and the b-value (2000) are suitable to obtain a good overall quality of the WM FODs, the spatial resolution was limited to 2.2 mm isotropic voxels. Since some bundles of white fibers are only a few mm wide, significant group differences in these bundles are difficult to detect at the resolution of the data used in the present study. This, however, was the maximum resolution that could be obtained for this signal while still maintaining a good signal-to-noise ratio.

7. Head motion: Previous investigations have shown the impact of head motion on estimated diffusion values. Even with a rigorous quality check and eddy current correction, head motion can significantly influence diffusion values. Have the authors assessed and controlled for head motion?

We agree that motion can significantly impact our results. We therefore excluded any individual that moved more than 2 mm from this study. In addition, we corrected for motion and described the used procedure in the dMRI processing section:

“All dMRI data were preprocessed using MRtrix3 (J. D. Tournier et al., 2019). Preprocessing steps included denoising (Veraart et al., 2016), Gibbs ringing correction (Kellner, Dhital, Kiselev, and Reisert, 2016), eddy-current and motion correction (Andersson and Sotiropoulos, 2016) and bias field correction (Tustison et al., 2010).”

8. On page: 9, it was not clear how the pairs of GM regions were defined/selected.

This was based on the significance of their connectivity. Using a matrix of regions, we selected the most significant gray matter regions as pairs. We have now clarified this further in the appropriate section:

“Using a matrix of regions, pairs of GM regions were made, based on the significance of their connectivity. These selected connectivity-based pairs of GM regions were used to extract the tracts connecting them, which allows to investigate the tract specific relations between GM volume, FD and FC and cognition. Using the template filtered tractogram (2M streamlines), we extracted the streamlines assigned to the pair of selected GM regions (Figure 1i). The resulting streamlines were then automatically thresholded into a binary fixel mask using an automated optimal threshold (Ridgway et al., 2009). The connectivity-based FC and FD values were then averaged in the mask.”

9. The imaging analyses controlled for age, sex and ICV but not for clinically relevant variables including disease duration, symptom severity, which limits the ability to interpret the results.

We appreciate the reviewer’s concern regarding this point. All these patients had the diagnosis of dementia due to FTD. Furthermore, all these patients had CDR = 1, and were still able to consent and to remain still within the scanning timeframe. A common metric of disease severity in such a clinically diverse population would be artificial and could potentially bias the results of the study. We recognize that the lack of this metric, imposes limitations on the interpretation of the results, and therefore now addressed this in the limitations of the study:

“Finally, although our imaging analyses controlled for age, sex and ICV but not for clinically relevant variables including disease duration and symptom severity, as these would be artificial and could potentially bias the results of a study with such a diverse clinical population, this imposes a limitation on the interpretation of the results presented in this study.”

10. Discussion: Finally, the discussion of vulnerability of the aslant tract to vascular pathology contributing to FTD pathology seems unsupported. This seems unlikely given that the majority of FTD patients are younger and less likely to have significant vascular disease. Could it be that the tract is vulnerable to any form of degenerative pathology?

We agree that there is a possibility for this to be the case. We would therefore like to thank the reviewer for this valuable contribution and added this as a separate hypothesis in the relevant section:

“Alternatively, given that the majority of FTD patients are younger and less likely to have significant vascular disease, it is possible that the tract is more vulnerable to a degree of degenerative pathology.”